# Seasonal effects of long-term warming on ecosystem function and bacterial diversity

**Melissa S. Shinfuku**[1], **Luiz A. Domeignoz-Horta**[1,2], **Mallory J. Choudoir**[1,3], **Serita D. Frey**[4], **Megan F. Mitchell**[5], **Ravi Ranjan**[6], **Kristen M. DeAngelis**[1]*

**1** Microbiology Department, University of Massachusetts, Amherst, MA, United States of America, **2** INRAE, AgroParisTech, UMR EcoSys, Université Paris-Saclay, Palaiseau, France, **3** Department of Plant and Microbial Biology, North Carolina State University, Raleigh, NC, United States of America, **4** Center for Soil Biogeochemistry and Microbial Ecology, Department of Natural Resources and the Environment, University of New Hampshire, Durham, NH, United States of America, **5** Graduate Program in Organismic and Evolutionary Biology, University of Massachusetts, Amherst, MA, United States of America, **6** Genomics Resource Laboratory, Institute for Applied Life Sciences, University of Massachusetts, Amherst, MA, United States of America

* deangelis@microbio.umass.edu

**Data Availability Statement:** The primer-trimmed bacterial sequencing data are available at the NCBI Sequence Read Archive under the BioProject accession number PRJNA957454. Ecosystem

## Abstract

Across biomes, soil biodiversity promotes ecosystem functions. However, whether this relationship will be maintained within ecosystems under climate change is uncertain. Here, using two long-term soil warming experiments, we investigated how warming affects the relationship between ecosystem functions and bacterial diversity across seasons, soil horizons, and warming duration. Soils were sampled from these warming experiments located at the Harvard Forest Long-Term Ecological Research (LTER) site, where soils had been heated +5°C above ambient for 13 or 28 years at the time of sampling. We assessed seven measurements representative of different ecosystem functions and nutrient pools. We also surveyed bacterial community diversity. We found that ecosystem function was significantly affected by season, with autumn samples having a higher intercept than summer samples in our model, suggesting a higher overall baseline of ecosystem function in the fall. The effect of warming on bacterial diversity was similarly affected by season, where warming in the summer was associated with decreased bacterial evenness in the organic horizon. Despite the decreased bacterial evenness in the warmed plots, we found that the relationship between ecosystem function and bacterial diversity was unaffected by warming or warming duration. Our findings highlight that season is a consistent driver of ecosystem function as well as a modulator of climate change effects on bacterial community evenness.

## Introduction

Climate change is driving losses in biodiversity, and these losses threaten both ecosystem productivity [1] and stability [2, 3]. While there is evidence that bacterial diversity promotes ecosystem function, not enough is known about how increased temperatures associated with climate change will affect this relationship. Studies examining the effects of warming on

function data (beta-glucosidase activity, N-acetyl glucosaminidase activity, oxidative enzyme activity, microbial biomass carbon, and respiration) is available in the Harvard Forest Data Archive under the Dataset ID HF431. Total soil organic carbon and nitrogen, z-score tranformed ecosystem function measurements, and ecosystem multifunctionality indices are available in the Supporting information and in the Harvard Forest Data Archive under the Dataset ID HF449.

**Funding:** This work was supported by a National Science Foundation (NSF, https://www.nsf.gov/) Long-Term Research in Environmental Biology grant (DEB-1456528) to SDF and KMD, and an NSF CAREER award (DEB-1749206) to KMD. The soil warming experiments at Harvard Forest are maintained with support from the National Science Foundation (NSF) Long Term Ecological Research Program (DEB-1832110). This work was also supported in part by a Graduate School Dissertation Research Grant from the University of Massachusetts Amherst to MSS (https://www.umass.edu/graduate). This research was also supported [in part] by the intramural research program of the U.S. Department of Agriculture (https://www.usda.gov), National Institute of Food and Agriculture, Hatch Multistate, accession number 7004345. The findings and conclusions in this publication have not been formally disseminated by the U. S. Department of Agriculture and should not be construed to represent any agency determination or policy. None of the funders had any role in study design, data collection and analysis, decision to publish, or preparation of the manuscript. There was no additional external funding received for this study.

**Competing interests:** The authors have declared that no competing interests exist.

microbial diversity and ecosystem function have revealed decreases in both microbial diversity and ecosystem function [4], a weaker relationship between microbial diversity and ecosystem function [5], or even abiotic modulators of the microbial diversity-ecosystem function relationship [6]. Increased temperatures also alter interactions between taxa in microbial communities [7, 8], which might explain why at higher temperatures decreases in diversity have disproportionate effects on ecosystem function [9]. Much of the basis for the current understanding of how warming affects the relationship between biodiversity and ecosystem function (BEF) has been gleaned from laboratory incubations or manipulated communities, which has limited our ability to accurately predict future ecosystem productivity. Clarifying whether the dual stresses of increased temperature and biodiversity loss affect ecosystem function in natural soils in the field is therefore critical to gaining a clearer understanding of the potential impacts of climate change.

Long-term global change experiments, like those at Harvard Forest Long-Term Ecological Research (LTER) site, provide a unique opportunity to examine long-term soil warming in a field setting. Given that the effects of warming on ecosystem function can change or compound over time [5, 10], maintaining and studying long-term climate change experiments is critical to the ability to predict ecosystem function. Additionally, microbial community compositional shifts resulting from climate change can take over a decade to manifest [11, 12]. How microbial communities respond to climate change is influenced by the duration of stress, with acclimation theorized to occur in the short-term and adaptation theorized to occur in the long-term [13]. Two soil warming experiments were established in 1991 and 2006 at the Harvard Forest, where soils have been continuously heated +5˚C above ambient along with control soils that have not been heated. Both the soil carbon quantity and quality has declined in response to heating [10], resulting in an increase in lipid concentration in the heated mineral soils [14]. The effects of warming on soil respiration at these sites has been nonlinear over decadal time scales at Harvard Forest [10], where there are years when there is a large treatment effect of heating compared to controls and years where there are no differences between heated and control plots. These changes in ecosystem function and nutrient pools have occurred in conjunction with changes to the microbial community. The abundance of Alphaproteobacteria as well as Acidobacteria have increased in the heated plots [11], and the overall diversity of the fungal community has decreased in the heated plots [7]. Altogether, these changes call into question how the relationship between ecosystem function and bacterial diversity will be altered by global warming.

Climate change alters both abiotic and biotic drivers of ecosystem function, which complicates predicting the response of ecosystem function to climate stresses, such as warming. One approach to addressing and condensing the dissimilar responses of individual ecosystem functions to climate change is the use of ecosystem multifunctionality (EMF) indices, which aggregate individual ecosystem functions into a single index. Compared to single function approaches, which examine ecosystem functions separately, EMF indices are more sensitive to biodiversity changes [15] and can better represent inherent trade-offs between ecosystem functions [16]. For instance in plant trait simulation study, ecosystem nitrogen use efficiency and litter quality had a negative relationship, indicating a trade-off between these two functions [17]. An EMF meta-analysis that incorporated results from 68 different warming studies observed that warming tends to have positive effects on EMF [18]. While kinetic effects are partially driving the positive effect of warming on EMF [19], warming-related changes in abiotic conditions, such as more limited substrate availability, also can increase certain ecosystem functions, like enzyme activity [14]. However, warming does not always have a positive effect on EMF, and when climate change alters abiotic conditions, warming can have a negative effect on EMF. For instance, warming decreases EMF under nitrogen limitation [20] or

precipitation declines [21]. Warming can also shift biotic drivers of EMF, like plant community functional traits, which in turn will have negative effects on EMF [22]. Microbial diversity and community composition, two of the most studied biotic drivers of EMF, also are influenced by warming, where decreases in dominant taxa due to warming can result in decreases in EMF [23]. The complex interactions between biotic and abiotic drivers of EMF have limited the integration of climate change into the existing framework of BEF relationships.

Temperate deciduous forests experience marked seasonal differences, and further complicating predictions, the effects of warming on ecosystem function and microbial diversity are also season-dependent. Long-term warming increases soil $CO_2$ efflux [10], and results in faster upticks in soil respiration after the spring thaw [24]. Additionally, warmed soils exhibit higher temperature sensitivity of respiration in the spring and winter [25], suggesting that with increasing temperatures, soil $CO_2$ efflux will increase non-linearly. Microbial biomass similarly has a seasonal cycle that tends to peak in fall [26, 27], although decreases in microbial biomass due to warming might alter this cycle [14]. Extracellular enzyme activity is likewise responsive to the seasonal shifts in the availability of carbon. Following or coinciding with leaf fall, the activity of peroxidase, phenol oxidase, and other oxidative enzymes involved in the decomposition of lignin tends to increase [27, 28]. Microbial diversity likewise exhibits seasonal shifts; however, the season with highest diversity varies from study to study. Summer, with its higher soil temperatures and higher availability of photosynthates as root exudates, has been observed as the season with the highest microbial diversity [29, 30]. Conversely winter has been noted as having the highest microbial diversity, due to the availability of products from the decomposition of fresh leaf litter in the fall [31]. Moreover, season is consistently identified as a driver of microbial community composition in forest soils [26, 32], where the community composition shifts along with leaf fall. While during the spring and summer, the community is dominated by organisms that can utilize root exudates, such as Proteobacteria, the fresh influx of leaf litter that occurs during the fall selects for organisms that can decompose lignin, such as Actinobacteria and *Mucilaginibacter* [30, 31, 33]. Despite variability in microbial communities and ecosystem functions across seasons, most studies that examine the relationship between ecosystem multifunctionality and microbial diversity utilize a single snapshot of the community and ecosystem at the time of sampling.

Here, we examined how 13 or 28 years of long-term soil warming affected ecosystem multifunctionality and bacterial diversity (Shannon's H), richness (Chao1), and evenness (Pielou's J) in two different seasons. We utilized previously published ecosystem function data [27], as well as additional ecosystem function data (soil total organic carbon and nitrogen) paired with bacterial 16S rRNA gene amplicon sequencing to test the hypothesis that warmed soils would have a stronger relationship between bacterial diversity and ecosystem multifunctionality than the control soils.

## Materials and methods

### Site description and sample collection

Soil samples were collected from two long-term warming soil experiments located at the Harvard Forest in Petersham, MA (42°30′30″N, 72°12′28″W), as described in [27]. The Prospect Hill Soil Warming Study was established in 1991 [34], and the Soil Warming x Nitrogen Addition (SWaN) Study was established in 2006 [24]. Warmed plots are heated continuously +5°C above ambient using buried resistance cables placed 10 cm below the soil surface and spaced 20 cm apart. Soil samples were collected in 2019 on July 15[th] and October 19[th], from Prospect Hill and SWaN, which had been warmed for 28 and 13 years, respectively, at the time of sampling. The experiments are located adjacent to one another and have the same dominant plant

overstory (*Acer rubrum, Betula lenta, Betula papyrifera, Fagus grandifolia, Quercus velutina, Quercus rubra*) and soil type (coarse-loamy incepitsols). The temperature ranges from a mean low of 3.3°C to mean high of 13.1°C [35]. The mean annual precipitation, including snow, is 1107 mm distributed evenly across seasons [35]. Both Prospect Hill and SWaN organic and mineral soils are acidic, with the organic horizon having pHs between 3.8 and 4.2 and the mineral soils having pHs between 3.9 and 4.4 [14, 36, 37]. The organic horizon in the control, unheated plots has a depth of approximately 5 cm, and the organic horizon in the heated +5°C plots has a depth of approximately 2–3 cm [14]. Duplicate cores were taken from each plot to 10 cm depth using a 5.7 cm diameter tulip bulb corer. Cores were separated into the organic and mineral horizons, duplicate soil cores were pooled by depth increment, roots and rocks were removed, and then the soil was sieved <2 mm. After sieving, samples were kept at ambient temperature, and within 4 hours of collection, samples were taken back to the lab for further analyses. The full sampling design was two sites (Prospect Hill, SWaN) x two treatments (control, heated) x two seasons (summer, fall) x two soil layers (organic horizon, mineral soil) x five replicate plots for a total of 78 samples (one sample from the SWaN experiment had a mineral horizon that was beyond the reach of the corer for the July sampling and this same plot was subsequently not sampled in October).

## Ecosystem functionality measurements

To assess ecosystem functionality, we utilized five soil properties or functions that were measured as a part of Domeignoz-Horta et al. (2023), including microbial biomass carbon, respiration, and the potential activities of four enzymes: phenol oxidase and peroxidase, β-glucosidase (BG), and N-acetyl-glucosaminidase (NAG). Additionally, we measured total soil organic carbon and total nitrogen. Altogether, there were seven different soil functions or properties measured for this study.

To measure total soil organic C and N and soil water content, soils were weighed into pre-weighed aluminum tins and dried in a 65°C oven until they reached a constant mass the same day the samples were brought back to the lab. Constant mass was verified by weighing the soils multiple times throughout the drying process. The drying temperature was selected so soils could be utilized for a separate analysis that was performed in Domeignoz-Horta et al. (2023) but not included in this study. After the soils achieved a constant mass, the tins were weighed again. The dry soil weight was subtracted from the wet soil weight, then divided by the dry soil weight to calculate water content. For total C and N, the dried soils were ground to a fine powder using a mortar and pestle. The soil was then weighed and packaged in duplicate into tins, which were run on a Perkin Elmer 2400 Series II CHN Elemental Analyzer with acetanilide as a standard at the University of New Hampshire Water Quality Analysis Lab. Total C and total N or duplicates were averaged.

Within three days of soil sampling, microbial biomass carbon was measured. Soils were stored at 15°C in the intervening time between sample collection and MBC measurement. The three day storage was to allow for similar treatment as samples used in carbon use efficiency calculations for Domeignoz-Horta et al. (2023). Four replicate soil samples were each split into two subsamples, with one group serving as a control and one group which was fumigated with chloroform vapors under vacuum pressure for 24 hours. Dissolved organic carbon (DOC) was then extracted from both the unfumigated and fumigated samples using 15 mls of 0.05 M $K_2SO_4$ and quantified on a Shimadzu TOC analyzer. Microbial biomass carbon was determined by subtracting the DOC concentration in the unfumigated subsample from the fumigated subsample.

Soils were stored at room temperature (20˚C) overnight before being aliquoted for soil respiration measurements. Soil respiration was measured on triplicate subsamples (0.15 or 0.3 g for the organic horizon or mineral soil, respectively) that were placed into Hungate tubes. Tubes were then sealed and incubated at 15˚C for 24 hours. A 30 ml headspace sample was then taken and injected into an infrared gas analyzer (Quantek 906) to measure $CO_2$ concentrations.

Prior to extracellular enzyme activity assays, soils were stored for no more than 4 days at 15˚C after soil sample collection. Potential extracellular enzyme activity was measured using fluorescent substrates. Soil slurries were prepared with 1.25 g wet weight soil and 175 mls of 50 mM pH 4.7 sodium acetate in a Waring blender. For the BG and NAG assays, 200 μls of soil slurry was pipetted into black 96 well plates, and for the oxidative enzyme assay, 500 μls of soil slurry was pipetted into deep well plates. Plates were then placed in a 15˚C incubator for 25 minutes to allow for temperature acclimation. This temperature reflects the average air temperature between the summer sampling and the fall sampling. After temperature acclimation, either 50 μls of 4000 μM 4-methylumbelliferyl β-D-glucopyranoside, or 50 μls of 2000 μM 4-methylumbelliferyl N-acetyl-glucosaminidase were added to each well. For assessing phenol oxidase and peroxidase activity, 500 μls of 25 μM L-DOPA + 0.03% $H_2O_2$ were added to each well. Each plate contained a standard curve as well as a slurry-only control. All plates were read on a SpectraMax M2 plate reader. BG and NAG plates were measured at 360/450 nm excitation/emission wavelengths after substrate addition. Oxidative enzyme plates were incubated for 4 hours after substrate addition, and then 100 μls were removed, transferred to a clear 96 well plate, and read at 460 nm. Since phenol oxidase and peroxidase both act on L-DOPA and 0.03% $H_2O_2$, any enzyme activity measured using these substrates was called oxidative enzyme activity. All enzyme activity was normalized for each sample by the sample's microbial biomass carbon.

## Ecosystem multifunctionality calculation

Multifunctionality was calculated for the organic horizons and mineral soil in R [38] using the multifunc package [39]. We included microbial biomass carbon, soil respiration, total carbon, total nitrogen, N-acetyl glucosaminidase activity, β-glucosidase activity, and oxidative enzyme activity in the ecosystem multifunctionality index. These functions represent both process rates (respiration, potential enzyme activity) and nutrient pools (total C, total N, microbial biomass) [40]. The organic horizon and mineral soil layers were analyzed separately due to documented differences in soil parameters [14] and microbiome properties [11, 41]. A multifunctionality index was calculated for each sample by taking the average of the z-score transformed ecosystem functions. While there are different ways of assessing diversity-multifunctionality relationships, we selected the averaging approach for ease of interpretation. Individual ecosystem functions' relationship with diversity was also investigated. For this approach, the raw untransformed function measurements were used.

## DNA extraction and library preparation

Bacterial diversity was measured using 16S ribosomal RNA (rRNA) gene amplicon sequencing. After sampling, soils were stored for three days at 15˚C to keep treatments consistent for an analysis in Domeignoz-Horta et al. (2023) that was not included in this study. After this three day storage period, soils were stored at -80˚C until extraction. DNA was extracted from soils using the Qiagen Powersoil kit following the manufacturer's protocol, and DNA concentration was measured using the Picogreen dsDNA kit (ThermoFisher Scientific).

We sequenced the 16S rRNA gene V4 region using the primers 515F (5′ – GTG YCA GCM GCC GCG GTA A– 3′) and 806R (5′ – GGA CTA CNV GGG TWT CTA AT – 3′) following the Earth Microbiome Project protocol [42], with minimal modifications. Forward primers contained the 5' Illumina adapter, 12 basepair unique Golay barcode, forward primer pad and linker, and 515F. Reverse primers contained the 3' reverse complement Illumina adapter, reverse primer pad and linker, and 806R. Templates were amplified in duplicate prior to sequencing using a 25 µl reaction containing 10 µM 2X Invitrogen Platinum Hot Start master mix, 0.5 µl 10 µM forward primer, 0.5 µl 10 µM reverse primer, 13 µl molecular grade $H_2O$, and 1 µl template. Samples were amplified in an Eppendorf Mastercycler Pro thermocycler with the program parameters of 94˚C for 3 min, 35 cycles of 94˚C for 45 s, 50˚C for 60 s, 72˚C for 90 s, and a final extension at 72˚C for 10 min. Technical replicate amplicons were combined and visualized on a 1% agarose gel. We quantified the amplicons with the PicoGreen Assay for dsDNA (ThermoScientific) and pooled 300 ng of each sample. We used AMPure XP magnetic beads (Beckman Coulter) to clean the pooled library. Prior to sequencing, the quality of the amplicon library was checked using Qubit Fluorometer (Thermo Fisher Scientific Inc) and 2100 Bioanalyzer DNA 7500 assay (Agilent Technologies, Inc). The library quantification was done using NEBNext Library Quant Kit for Illumina (New England Biolabs). The amplicon library was spiked with Illumina PhiX and sequenced on Illumina MiSeq platform using the v2–300 cycle kit with 156 bp paired end chemistry (Illumina Inc). Custom Read1, Read2, and Index Sequencing primers were used for sequencing. Negative controls with no template were PCR amplified and sequenced as well. Amplicon library quality assessment and sequencing was performed at the Genomics Resource Laboratory (RRID:SCR_017907), Institute for Applied Life Sciences, University of Massachusetts Amherst, MA, USA.

We were also interested in measuring fungal community diversity using ITS amplicon sequencing. However, we were unable to successfully PCR amplify enough samples to have triplicates for each combination of season, soil layer, warming treatment, and warming duration. Thus, the focus of this study considered only on the bacterial community.

## Sequencing processing and analysis

Raw FastQ files were demultiplexed and primers trimmed using cutadapt [43]. Bacterial 16S rRNA gene sequences were processed with the DADA2 pipeline (v.1.18.0) [44], which generates amplicon sequence variants (ASV), in R. Forward reads were trimmed to 150 bp, and reverse reads were trimmed to 140 bp. Any reads with expected error higher than 2 or with any ambiguous nucleotides were discarded, and all PhiX sequences were also removed. Error rates for the forward and reverse reads were estimated separately using the function 'learnErrors'. ASVs were called for the forward and reverse reads using their respective error models. Denoised forward and reverse ASVs were merged using the function 'mergePairs' with default parameters. Prior to assigning taxonomy, any chimera sequences were identified *de novo* and removed. Taxonomy was assigned using SILVA (v 138.1) [45]. All sequences that were classified at the family level as "mitochondria" or at the order level as "chloroplast" were filtered out from the final ASV table. To account for differences in sequencing depth between samples, raw ASV counts were normalized with library size estimation factors calculated in DESeq2 [46, 47]. Any ASVs that had 1 or 0 reads total across all samples after normalization were discarded.

Our final data set comprised 69 out of 79 possible samples due to difficulty with successfully PCR amplifying samples during library preparation. In the final data set, each combination of season, soil layer, warming treatment, and warming duration had at least three replicates out of the original five replicates, except the October organic horizon control treatment from the

13-year experiment, which had only two out of five replicates. At the end of sequence processing, the data set had 4,345,430 reads, with an average depth of 62,977 reads per sample, and 8,754 total ASVs. The no template negative controls had fewer than 500 reads. After correcting with a sample specific library size estimate calculated in DESeq2 [46], the total read count was 4,159,442, with an average read count of 60,281 per sample. Rarefaction curve analyses indicated that for the samples that were successfully sequenced, the sequencing efforts were adequate.

Since a variety of diversity metrics have been correlated with EMF, we calculated Simpson diversity, Shannon's H, Chao1 estimated richness, community structure (PCoA axes), and evenness (Pielou's J). Diversity metrics were calculated using the packages vegan [48] and phyloseq [49]. The distance matrix for the principal coordinate analysis was calculated using Bray-Curtis dissimilarity using the function 'vegdist' in the package vegan.

## Statistical analysis

All statistical analyses were done in R [38]. Prior to any statistical analysis, we evaluated whether warming treatment and warming duration could be combined into one variable, where a single "control" treatment (or "0 year warmed") would derive from combined Prospect Hill and SWaN plot control samples, Prospect Hill warmed samples would be "28 year warmed", and SWaN warmed samples would be "13 year warmed". Both EMF and bacterial diversity metrics were tested for normality using a Shapiro-Wilk test. After checking for normality, we tested for significant differences between the control plot samples from Prospect Hill and SWaN using either a Wilcoxon rank sum test if the data were not normally distributed (as was the case for both the organic horizon and mineral soils' EMF), or a t-test if the data were normally distributed (as was the case for the organic horizon and mineral soils' bacterial diversity metrics). We found no significant differences between the control plots' EMF in Prospect Hill and SWaN. However, we did find significant differences in bacterial diversity between the control plots from the two experiments (p = 0.0260). As a result, warming treatment and warming duration were combined into a single variable when EMF was the response variable, but not when bacterial diversity was the response variable.

We also investigated whether percent water content correlated with ecosystem multifunctionality or any of the bacterial diversity metrics in the organic horizon or mineral soils. Prior to correlation testing, ecosystem multifunctionality, Shannon H, Chao1 estimated richness, and Pielou J were tested for normality using a Shapiro-Wilk test. Of these four variables, the organic horizon bacterial diversity metrics and mineral soil Shannon's H were normally distributed. All other variables were not normally distributed. For variables that were normally distributed, we used a Pearson correlation test. For variables that were not normally distributed, we used a Spearman correlation test.

We built a set of general linear models using raw or log-transformed data to investigate the effects of warming, soil layer, season, and/or site on four response variables: individual ecosystem functions, nutrient pools, EMF, or bacterial diversity. When normality of model residuals was not satisfied or the response variable was not normally distributed, we constructed generalized linear models with non-Gaussian distributions. We considered both additive and interactive effects between warming, soil layer, season, and/or site.

A set of candidate generalized linear models were constructed with either EMF or bacterial diversity as the response variable. First, models were built using a Gaussian distribution and model residuals were tested for normality, except for models where Chao1 estimated richness was the response variable. When Chao1 estimated richness was the response variable, models utilized a Poisson or negative binomial distribution, which are both appropriate

for count data. Poisson models for Chao1 estimated richness were checked for overdispersion by looking at the ratio between residual deviance and degrees of freedom, and if overdispersion was observed (ratio > 1.1), a negative binomial distribution was used with the R package MASS [50]. For the mineral soils, any model where EMF was the response variable had a log-transformation applied to EMF to make the residuals normally distributed. The results from the log-transformed models are reported without back-transformation. We found that the different diversity metrics all performed similarly based on their ΔAICc, so we elected to utilized Shannon's H, Pielou's J, and Chao1 estimated richness as predictors since taxon richness and community evenness have both been found to be drivers of ecosystem function [51, 52].

We also tested whether warming treatment, warming duration, soil layer, and/or season affected the relationship between the individual ecosystem functions or nutrient pools and bacterial diversity by constructing linear models. Model residuals were tested for normality using a Shapiro-Wilk test. If residuals were not normally distributed, a log-transformation was applied. If the model residuals were still not normally distributed after a log-transformation, we constructed a set of generalized linear models using a gamma or inverse Gaussian distribution. Residuals for these models were validated using the DHARMa package [53]. Results for models where a log-transformation was applied or for models that did not use a normal distribution are reported without back-transformation.

Finally, we then investigated how warming treatment, warming duration, soil layer, and/or season affected the relationship between EMF and bacterial diversity. For all of these analyses, we constructed a set of candidate linear models. In these sets of candidate models, we considered all predictors and possible interactions between warming treatment and diversity, warming treatment and horizon, warming treatment and season, or warming treatment and warming duration. All model residuals were checked for normality, and if necessary a log-transformation was applied to the response variable. Estimates and errors from the log-transformed models are reported without back-transformation.

For all of the candidate sets of models, model fit was assessed using AICc in the package [54]. If models were within 2 ΔAICc units, we considered the models equally supported, and used a likelihood ratio test for the nested models to select a model. If the models with similar ΔAICc were not nested, we selected the model that explained more variance in the data. Model residual deviance was also compared to model null deviance to examine model fit. When significant interactions were found between warming treatment, warming duration, and/or season, a Tukey's HSD test was utilized to further investigate the interaction. Otherwise, p-values had a Benjamini-Hochberg correction applied to them to account for multiple comparisons. We utilized a p-value cutoff of 0.05 for statistical significance. If the p-values were between 0.05 and 0.1, we report it as a trend. All estimates are reported with a 95% confidence interval.

After analyzing the bacterial diversity and finding a significant difference in bacterial evenness between the warmed and control treatments in the summer organic horizon samples, we conducted a *post-hoc* analysis to see whether any of the dominant ASVs were differentially abundant between control and heated plots. We used three different methods: Analysis of Community Microbiomes with Bias Correction in the R package ANCOM-BC (v.2.0.3) [55], log2 fold change using the package DESeq2 [46], and a $\chi^2$ test between ASV counts in the heated and control plots. Dominance was determined based on a relative abundance threshold of ≥ 0.001 within each sample. Of the 8,629 ASVs in the organic horizon summer dataset, 256 ASVs were identified as dominant, which accounted for 2.967% of the total ASVs. For all three differential abundance methods, p-values were adjusted for multiple comparisons using a Benjamini-Hochberg correction.

**Table 1. The effects of warming treatment, season, or site on diversity metrics in the organic horizon and mineral soil.**

| soil layer | diversity metric | predictor | estimate | error | p value |
|---|---|---|---|---|---|
| Organic horizon | Shannon diversity | season (15) | -0.076 | 0.157 | n.s. |
| | | warming (16) | -0.134 | 0.151 | n.s. |
| | | warming:season (9) | 0.247 | 0.220 | 0.072 |
| | Chao1 estimated richness | site (16) | -0.110 | 0.086 | 0.019 * |
| | | warming (16) | 0.038 | 0.0.86 | n.s. |
| | Pielou evenness | season (15) | -0.011 | 0.016 | n.s. |
| | | warming (16) | -0.024 | 0.016 | 0.011 * |
| | | warming:season (9) | 0.031 | 0.024 | 0.015 * |
| Mineral soil | Shannon diversity | season (19) | 0.011 | 0.151 | n.s. |
| | | warming (19) | 0.001 | 0.015 | n.s. |
| | Chao1 estimated richness | site (19) | -0.047 | 0.122 | n.s. |
| | | warming (19) | 0.017 | 0.122 | n.s. |
| | Pielou evenness | site (19) | -0.006 | 0.022 | n.s. |
| | | warming (19) | 0.002 | 0.022 | n.s. |

Shannon diversity models and organic horizon Pielou evenness models used a Gaussian distribution, Chao1 estimated richness models used a negative binomial distribution, and mineral soil Pielou evenness models used a gamma distribution. Chao1 estimated richness and mineral soil Pielou evenness estimates are reported in the table without back-transformation. Estimates are reported where the reference for warming treatment is control non-heated plots, the reference for season is summer, and the reference for site is the 13 year experiment. Number of samples for the contrasts are reported in parentheses. Errors are reported for a 95% confidence interval. Any significant interactions were further investigated using a Tukey HSD test. Benjamini-Hochberg adjusted p-values are reported, and any adjusted p-values that were greater than 0.05 are reported as not significant (n.s.).

## Results

### Bacterial diversity

The effect of warming on bacterial evenness varied by season in the organic horizon, where the best fitting model based on ΔAICc was the interaction model with warming and season (Table 1). Warming had a negative impact on Pielou's J in the summer but no effect in the fall (Tukey HSD summer: -0.024 ± 0.011, p = 0.026; Tukey HSD fall: p = 0.801). This pattern was not observed in the mineral soil, where warming did not affect bacterial evenness. None of the bacterial diversity metrics in the organic horizon had a significant correlation with soil water content (S1 Table). There was also a season-dependent warming trend on Shannon diversity, although this effect was not significant (p = 0.072). Since in the organic horizon in the summer warming decreased bacterial evenness but did not affect Chao1 richness, we carried out a *post-hoc* differential abundance analysis on dominant ASVs to determine if changes in the abundance of dominant ASVs could explain the decrease in evenness. None of the dominant ASVs were differentially abundant between the warmed and control plots.

Chao1 estimated richness significantly differed between the two different sites, Prospect Hill (28 year warmed) and SWaN (13 year warmed) in the organic horizon. The older experimental site had around 90% of the number of taxa compared to the younger site (Table 1). Soil water content did not have a significant correlation with any of the bacterial diversity metrics in the mineral soils (S1 Table). Unheated control plots from the 13-year warmed and 28-year warmed experiments had significantly different bacterial diversity metrics and so were not combined when bacterial diversity was the response variable.

## Ecosystem multifunctionality

Ecosystem multifunctionality was affected only by season and not warming treatment or warming duration. In the organic horizon, EMF was significantly higher in fall compared to summer (0.089 ± 0.064, p = 0.022, Fig 1A). Similarly, in the mineral soil, EMF trended higher in fall compared to summer, although this difference was not statistically significant (p = 0.090, Fig 1B). Neither EMF in the organic horizon nor EMF in the mineral soils had a significant correlation with soil water content (S1 Table). Due to known differences between the organic horizon and mineral soil [14], ecosystem multifunctionality was calculated for the soils separately.

For both the organic horizon and mineral soils, neither 13 years nor 28 years of warming significantly altered EMF. In mineral soils, EMF tended to increase as warming duration

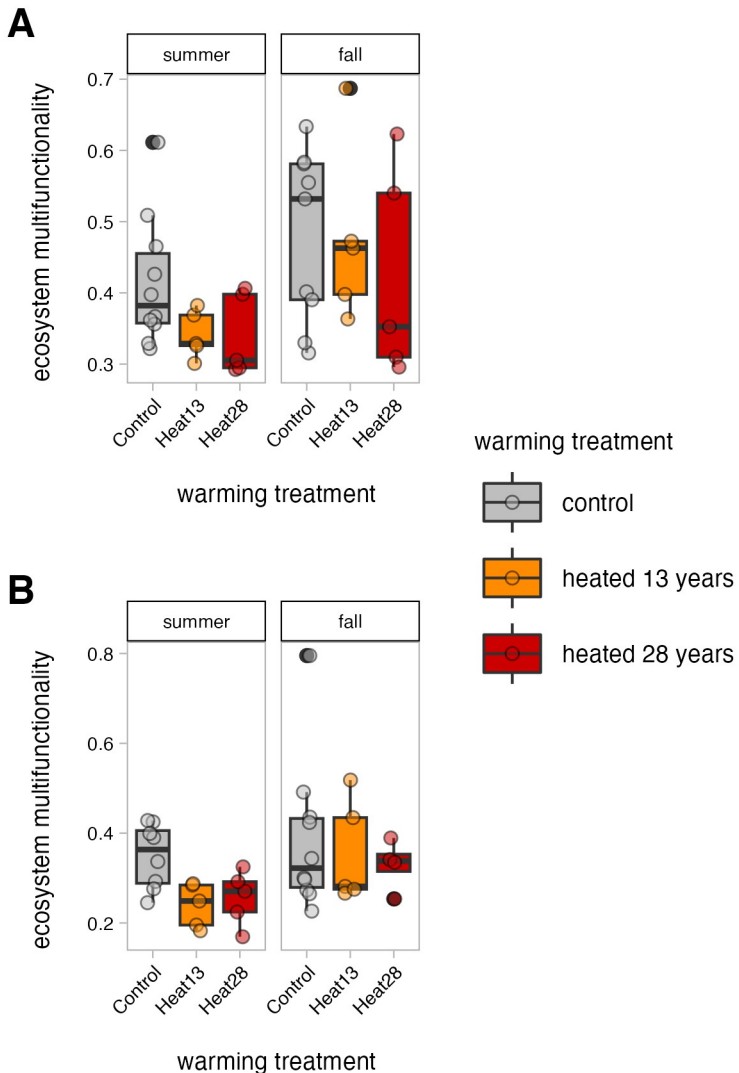

**Fig 1. Ecosystem multifunctionality across warming treatment and season.** Ecosystem multifunctionality (EMF) was calculated by averaging z-score standardized ecosystem function measurements for each sample, and EMF was calculated separately for the organic horizon (A) and mineral soils (B). In the mineral soils (B), EMF was log-transformed for general linear model construction, but original non-log-transformed values are displayed.

increased, but these differences were not significant. In the organic horizon soils, we did not observe any trend in EMF with longer duration of heating.

### Ecosystem multifunctionality-bacterial diversity relationship

In the organic horizon, season had a significant effect on the relationship between EMF and bacterial Chao1 estimated richness, where fall had significantly higher intercept for EMF than summer, irrespective of warming treatment or duration (Fig 2A, 0.108 ± 0.074, p = 0.020). We

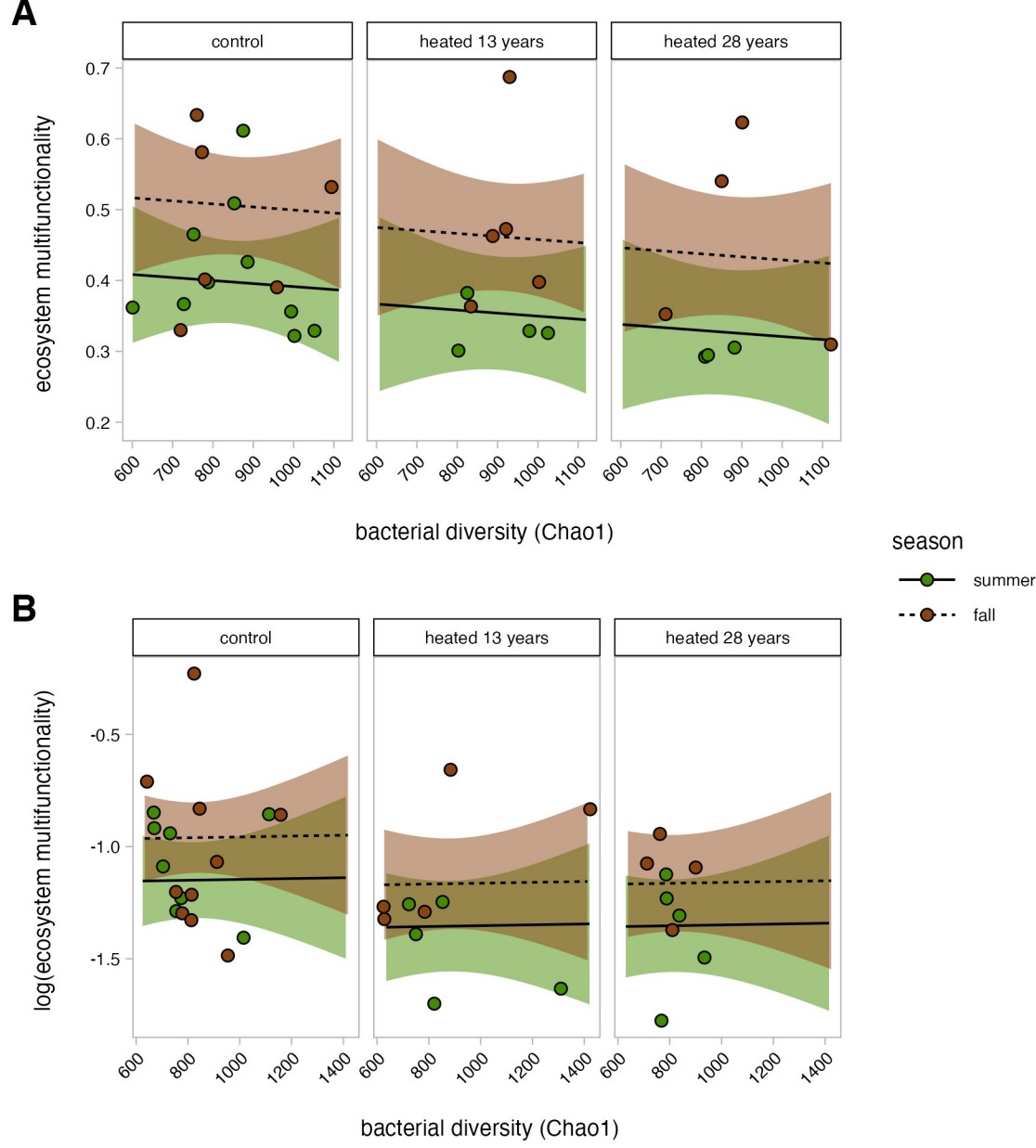

**Fig 2. The relationship between ecosystem multifunctionality and bacterial Chao1 estimated richness.** Ecosystem multifunctionality (EMF) was calculated by averaging z-score standardized function measurements for each sample. Shaded regions represent a 95% confidence interval. Mineral soil EMF was log-transformed and is displayed without back-transformation.

observed a similar, seasonal trend in the relationship between EMF and Shannon's diversity (S2A Fig, 0.105 ± 0.075, p = 0.052) and Pielou's evenness (S2C Fig, 0.104 ± 0.074, p = 0.055). In the organic horizon soils, ecosystem multifunctionality did not correlate with bacterial Chao1 estimated richness, Shannon diversity, or Pielou evenness. The relationship between EMF and Chao1 estimated richness had a progressively lower intercept as warming duration increased. However, neither the 13-year warmed nor the 28-year warmed plots showed a relationship between EMF and Chao1 estimated richness that was significantly different from the control plots.

In the mineral horizon, Shannon diversity (S2B Fig), Chao1 estimated richness (Fig 2B), or Pielou evenness (S2D Fig) were not correlated with EMF, and the intercept for the relationship between EMF and Chao1 estimated richness did not change as warming duration increased. There also was no effect of season on the intercept for EMF for all three of the diversity metrics in the mineral soil.

## Individual ecosystem functions-bacterial diversity relationship

In the organic horizon, no individual ecosystem function or nutrient pool had a significant relationship with any diversity metric. MBC and oxidative enzyme activity both varied significantly by season, with MBC higher in fall compared to summer and oxidative enzyme activity higher in fall compared to summer for both the Shannon diversity and Chao1 estimated richness models (Table 2 and S2 Table).

In the mineral soil, β-glucosidase activity was the only ecosystem function that had a relationship with any bacterial diversity metric, where Shannon's H had a negative relationship with β-glucosidase activity (Table 3 and S3 Table). Similar to the organic horizon, significant seasonal differences were observed for MBC, oxidative enzyme activity, and β-glucosidase

**Table 2. Single ecosystem function-diversity relationships in the organic horizon.**

| ecosystem function | diversity metric | predictor | estimate | error | p value |
|---|---|---|---|---|---|
| MBC | Shannon diversity | season (15) | 2149.600 | 537.432 | 9.942e-08 *** |
| | | 13 years heating (9) | -230.400 | 622.104 | n.s. |
| | | 28 years heating (7) | -470.800 | 675.612 | n.s. |
| | | Shannon (32) | 341.100 | 1654.436 | n.s. |
| | Chao1 estimated richness | season | 2165.129 | 533.296 | 7.456e-08 *** |
| | | 13 years heating (9) | -246.062 | 636.418 | n.s. |
| | | 28 years heating (7) | -477.112 | 678.174 | n.s. |
| | | Chao1 (32) | 0.1276 | 2.248 | n.s. |
| OX activity | Shannon diversity | season | -11573.7 | 1724.212 | 1.468e-12 *** |
| | | 13 years heating (9) | 122.400 | 231.476 | n.s. |
| | | 28 years heating (7) | 101.300 | 233.828 | n.s. |
| | | Shannon (32) | 245.900 | 545.664 | n.s. |
| | Chao1 estimated richness | season | -1.157e+04 | 1765.960 | 2.579e-12 *** |
| | | 13 years heating (9) | 1.419e+02 | 229.712 | n.s. |
| | | 28 years heating (7) | 1.225e+02 | 229.908 | n.s. |
| | | Chao1 (32) | 0.186 | 0.706 | n.s. |

Microbial biomass (MBC) models used a Gaussian distribution and oxidative enzyme (OX) models used an inverse Gaussian distribution. Oxidative enzyme model estimates are reported without back-transformation. All model estimates are reported where the reference for warming treatment is control non-heated plots and the reference for season is summer. Number of samples are reported in parentheses. Errors are reported for a 95% confidence interval. Benjamini-Hochberg adjusted p-values are reported. Any adjusted p-values greater than 0.05 are reported as not significant (n.s.).

**Table 3. Single ecosystem function-diversity relationships in the mineral soil.**

| ecosystem function | diversity metric | predictor | estimate | error | p value |
|---|---|---|---|---|---|
| MBC | Shannon diversity | season (19) | 0.944 | 0.278 | 8.355e-07 *** |
| | | 13 years heating (10) | -0.294 | 0.333 | n.s. |
| | | 28 years heating (9) | -0.318 | 0.345 | n.s. |
| | | Shannon (37) | 0.004 | 0.623 | n.s. |
| | Chao1 estimated richness | season (19) | 0.942 | 0.276 | 3.833e-07 *** |
| | | 13 years heating (10) | -0.309 | 0.333 | n.s. |
| | | 28 years heating (9) | -0.313 | 0.343 | n.s. |
| | | Chao1 (37) | 0.0002 | 0.0008 | n.s. |
| OX activity | Shannon diversity | season (19) | 1.927 | 0.247 | 1.479e-15 *** |
| | | 13 years heating (10) | -0.329 | 0.296 | n.s. |
| | | 28 years heating (9) | -0.320 | 0.308 | n.s. |
| | | Shannon (37) | -0.024 | 0.555 | n.s. |
| | Chao1 estimated richness | season (19) | 1.926 | 0.247 | 7.401e-16 *** |
| | | 13 years heating (10) | -0.334 | 0.298 | 0.058 |
| | | 28 years heating (9) | -0.318 | 0..298 | 0.063 |
| | | Chao1 (37) | 8.007e-05 | 7.207e-04 | n.s. |
| BG activity | Shannon diversity | season (19) | -744.41 | 216.306 | 6.389e-07 *** |
| | | 13 years heating (10) | -96.970 | 258.720 | n.s. |
| | | 28 years heating (9) | -157.880 | 268.598 | n.s. |
| | | Shannon (37) | -623.500 | 482.866 | 0.028 * |
| | Chao1 estimated richness | season (19) | -747.101 | 232.613 | 2.316e-06 *** |
| | | 13 years heating (10) | -91.320 | 280.160 | n.s. |
| | | 28 years heating (9) | -150.038 | 288.798 | n.s. |
| | | Chao1 (37) | -0.375 | 0.678 | n.s. |

A log-transformation was applied to microbial biomass carbon (MBC) and oxidative enzyme (OX) activity. MBC models, OX models, and β-glucosidase (BG) models all used a Gaussian distribution. MBC model estimates and OX model estimates are reported without back-transformation. All model estimates are reported where the reference for warming treatment is control non-heated plots and the reference for season is summer. Number of samples for contrasts/predictor are reported in parentheses. Errors are reported for a 95% confidence interval. Benjamini-Hochberg adjusted p-values are reported. Any adjusted p-values greater than 0.05 are reported as not significant (n.s.).

activity. Both MBC and oxidative enzyme activity were higher in fall compared to summer, whereas the opposite was observed for β-glucosidase activity, irrespective of the diversity metric (Table 3).

## Discussion

Season is a critical driver of ecosystem function, especially in deciduous forests. We observed that ecosystem multifunctionality was significantly higher in fall compared to summer. Indeed, season had a larger effect on EMF than warming treatment, which had no effect on EMF. The seasonal input of fresh litter could explain the difference in EMF between fall and summer. With leaf-fall, a priming effect could be initiated, resulting in higher microbial activity [27, 56, 57]. This priming could be substantial enough to mitigate the negative impact of warming on EMF. Further supporting a priming effect, we observed a significant increase in microbial biomass in fall, and an increased microbial biomass has also been noted following the addition of fresh carbon in priming experiments [58–60]. In the heated plots at Harvard Forest, soil organic matter is more depleted in simple sugars and is lower in quantity compared to soil

organic matter in the control plots [61], and the influx of litter in the fall might help alleviate this resource limitation [27]. Accordingly, seasonal influx of nutrients cancels out the warming-induced substrate limitation which can be observed by the response of the microbial biomass carbon [27]. This could subsequently increase enzyme production and activity and microbial biomass, both of which were included in our EMF index. While we only examined two seasons, summer and fall, our results highlight the importance of accounting for seasonal differences when measuring ecosystem function.

An additional contributor to the higher EMF observed in fall could be a two-day rainfall event that occurred two days prior to the fall sampling [35]. This rainfall could have initiated a Birch effect, which is an increase in respiration that occurs after rewetting a dried soil [62]. Furthermore, the rainfall could have resulted in a flush of dissolved organic matter, which was found in Domeignoz-Horta et al. (2023). Drying-rewetting cycles can increase microbial biomass [63, 64], which is in line with the significantly higher microbial biomass that we observed in fall (Tables 2 and 3). Warmed soils have significantly lower soil moisture in the spring and fall compared to the control plots in the [24], and the combination of warming and lower moisture can further increase $CO_2$ efflux after rewetting [65]. Yet, we did not observe a significant increase in respiration in fall [27], as would be expected with a Birch effect. This could be due to the fact that respiration was measured after the soil was taken back to the lab, up to six days after the rainfall. This falls outside of the 1.5 days when the respiration spike that is characteristic of the Birch effect is typically observed [66]. Enzyme activity can also increase after rewetting [67]. While we did observe increased phenol oxidase activity in the fall, we did not observe similar increases in activity for β-glucosidase and N-acetyl glucosaminidase [27]. Finally, we did not observe any significant correlation between soil water content and EMF (S1 Table). Even though we noted higher EMF in the fall, the ecosystem function responses we observed are not entirely consistent with a rewetting response.

Bacterial community evenness tends to be more sensitive to warming compared to bacterial community richness. We observed that bacterial community evenness was negatively affected by soil warming and that this effect was specific to the organic horizon in the summer (Table 1). This contrasts with previous work at our site, where warming increased bacterial community evenness in the organic horizon as a result of decreases in dominant taxa abundance [11]. Warming often drives shifts in community evenness, whether through increased dominance of a single taxon, decreases in dominant taxa, or losses of rare taxa [7, 68, 69]. We did not observe any significant effects of warming on bacterial community richness (Table 1), which is in alignment with other experimental warming studies or meta-analyses [18, 70–72]. Changes in abundance of dominant taxa were not the drivers of this decrease in evenness either.

Changes in substrate quality and quantity due to warming might also explain the observed changes in evenness. At the Harvard Forest, warming has negatively affected both substrate quality and quantity [27], and lower substrate availability likewise has negative effects on microbial community evenness [73]. Given that ecosystem function tends to be more resilient to stress in more even communities [52], the negative effects of warming on evenness ultimately could make ecosystem function less stable. Yet these negative effects might be masked or compensated for by changes to microbial community function as a result of heating, such as increased abundance of carbohydrate activated enzymes (i.e. CAZymes) involved in degrading complex carbohydrates [41] or an expanded number of bacteria able to degrade lignin [74].

In this study, we observed no significant relationship between bacterial diversity, richness, or evenness and EMF, irrespective of season, soil horizon, warming treatment, or warming duration. Typically, ecosystem multifunctionality positively correlates with bacterial diversity [75–77], but neutral, or marginally positive relationships are not uncommon [78, 79]. In fact,

abiotic or environmental factors, such as soil moisture or pH, are consistently identified as stronger drivers of EMF than diversity itself [51, 79–81]. Even the spatial scale of sampling can affect the relationship between EMF and diversity, where at larger scales, like continental or global scales, diversity and ecosystem function tend to be positively correlated, but at smaller scales, like plot level or regional scales, diversity and ecosystem function tend to be neutrally or negatively correlated [82]. Additionally, active microbial diversity [83] or functional trait diversity [84–86] may be a better predictor of ecosystem multifunctionality than total or taxonomic microbial diversity. This may explain why we did not see a relationship between bacterial diversity and EMF. Our DNA-based sequencing approach captured total taxonomic diversity, which comprises both the active and inactive community, and did not provide any functional trait information. Moreover, in soils many bacteria are dormant, with minimal metabolic activity [87], and these dormant cells are not likely contributing to EMF. Altogether, this suggests that when microbial diversity positively correlates with ecosystem function, this correlation tends to be weak, mainly driven by active members of the community or specific functional groups, and ultimately can be better explained by abiotic variables.

Functional redundancy within the bacterial community can also account for the lack of relationship between EMF and bacterial diversity. In theory, if an ecosystem function is functionally redundant within a community, the addition of more taxa that carry out that specific function will have negligible effects on the performance of that function at any given time. Forest soils are known to contain high levels of functional redundancy [88], therefore changes in diversity might not result in changes in function. Additionally, the functions included in the ecosystem multifunctionality index we calculated are known to be widespread within soil microbial communities. For instance, aerobic respiration is widespread throughout the phylogenetic tree, and the genes for β-glucosidase or N-acetyl glucosaminidase production are found or predicted in Proteobacteria, Chloroflexi, and Actinobacteria [89, 90], all of which were abundant in our samples. Conversely, ecosystem functions that are performed by a narrower group of taxa and therefore exhibit less functional redundancy, such as xenobiotic degradation or nitrogen fixation, are more sensitive to changes in diversity [80, 91]. Although, even within functions that are considered to be less widespread throughout the phylogenetic tree, such as nitrite oxidation, there does seem to be some level of functional redundancy or resilience in response to changes in diversity [92]. The lack of a relationship between bacterial diversity, richness, or evenness and EMF could partially be attributed to the functional redundancy of the ecosystem functions in this study.

In these temperate terrestrial forest soils, fungi could have a large role in ecosystem function, which would explain why we did not observe a relationship between any bacterial diversity metric and ecosystem multifunctionality. Both bacterial and fungal diversity are known drivers of ecosystem function [51, 77], and ecosystem multifunctionality-microbial diversity studies rely on the assumption that bacteria and fungi are equally involved in promoting ecosystem functions. While we selected our seven ecosystem functions as representatives of processes such as carbon cycling rates (β-glucosidase activity, respiration, oxidative enzyme activity), nitrogen cycling rates (N-acetyl glucosaminidase activity), or nutrient pools (total carbon, total nitrogen, microbial biomass), some of these functions might be more dependent on fungi. For instance, fungal biomass typically outnumbers bacterial biomass in temperate forests [93], although previous work at the sites in this study found that bacteria outnumbered fungi, based on 16S rRNA or ITS gene copy numbers [11]. Enzyme production and activity could also be skewed towards fungi since fungi are the main producers of oxidative enzymes, like phenol oxidase, in soils [94]. N-acetyl glucosaminidase targets chitin, which is a part of the fungal cell wall, and NAG activity is correlated with fungal biomass as well [95]. While we did initially set out to examine both the bacterial and fungal communities, we were unable to

successfully PCR amplify the fungal community during sequencing library preparation. As a result, we could not assess the role of the fungal community in driving ecosystem multifunctionality. Even though bacteria play a role in ecosystem function, fungi are still major contributors to EMF, possibly explaining why we did not observe a relationship between EMF and any bacterial diversity metric.

While we did not observe a relationship between EMF and bacterial diversity, richness, or evenness, we did observe that β-glucosidase activity had a significant relationship with bacterial Shannon diversity in the mineral soils. Unexpectedly, this relationship was negative (Table 3), meaning that soils with higher diversity tended to have lower β-glucosidase activity potential. We only observed this negative relationship for bacterial Shannon diversity and not bacterial Chao1 richness, suggesting that more even bacterial communities have lower BG activity. This could be explained by negative selection effects, which arise when the most abundant or dominant taxa do not perform or add to a specific ecosystem function, making the diversity-ecosystem function relationship negative or neutral [96]. In fact, β-glucosidase activity can be driven by individual taxa that are not the most abundant within a community [97], and the energetic costs of extracellular enzyme production might negatively impact growth of extracellular enzyme producers [98]. As a result, the presence or absence of individual taxa that are associated with high BG activity might not be reflected in diversity metrics that assess total community diversity. Further, metabolic overlap, which occurs when different taxa within a community can utilize the same substrate or compound, might also account for the negative relationship between diversity and function, since more diverse communities tend to have less metabolic overlap and more diverse metabolic pathways [99]. As it relates to β-glucosidase activity, higher diversity communities might have more diverse metabolisms that can utilize other substrates besides glucosyl, the substrate of BG, and therefore might have lower overall BG activity. Negative selection effects and metabolic overlap can both account for the negative relationship between bacterial diversity and β-glucosidase activity, which has also been observed in other studies [23, 100].

Controlled manipulative laboratory incubations and observational field studies are complementary methods to understand the drivers of the BEF relationship, and both of these methods have validated the relationship between ecosystem function and microbial diversity. Mycorrhizal and saprotrophic fungal diversity and bacterial diversity were all noted as having positive correlations with ecosystem multifunctionality in a study that sampled from 80 different sites across the globe [51]. Manipulative laboratory experiments, where diversity is altered via serial dilutions, have also observed a similar positive trend [101, 102]. However, the strength of the positive correlation tends to decrease when comparing manipulative studies to observational studies, due to factors such as lack of immigration and dispersal, reduced environmental complexity, or short experimental duration [84]. Additional variability in the strength of the relationship between microbial diversity and ecosystem function arises from abiotic factors, like season or soil type [79]. Indeed, we observed no relationship between bacterial diversity and ecosystem function in this observational field study. However, the removal of abiotic variables can affect how diversity confers benefits to ecosystem function. The benefits of diversity arise from mechanisms such as niche partitioning [5] or maintaining a pool of taxa that can thrive under different environmental conditions [52, 103, 104]. Niche partitioning, in particular, is dependent on the presence of environmental heterogeneity in order for organisms to colonize the niche they are best adapted to [105]. Further emphasizing the importance of abiotic variables, we found that season had the largest impact on ecosystem function. Thus, while controlled laboratory experiments have helped to elucidate the mechanisms driving the BEF relationship, observational field studies are still necessary to fully understand the role that abiotic factors, such as season, have on this relationship under more natural conditions.

## Conclusion

This study investigated if long-term soil warming influenced the relationship between ecosystem multifunctionality and bacterial diversity. We observed a strong seasonal effect on ecosystem function, where ecosystem multifunctionality was significantly higher in the fall than in the summer. The effects of soil warming on bacterial evenness were also dependent on season in the organic horizon, where in the summer warming negatively affected bacterial evenness, but in the fall warming did not affect bacterial evenness. However, we found that warming treatment or warming duration ultimately had no effect on the relationship between EMF and bacterial diversity. Indeed, we found that EMF had no relationship with bacterial diversity, which is not uncommon in natural or non-laboratory systems. The younger experimental site also had increased bacterial richness in the organic horizon. Overall, we found that season exerts a strong influence on ecosystem functionality, and season modulates the effects of warming in temperate deciduous forests. We emphasize that cross-season sampling is needed to best assess an ecosystem's total function and that diversity metrics other than taxonomic diversity may be better suited to capturing the role of diversity on ecosystem function.

## Supporting information

**S1 Fig. Relative abundance of the phyla of dominant bacteria based on 16S rRNA gene amplicon sequencing.** Taxonomy was assigned using SILVA (v 138.1). Sample amplicon sequence variant (ASV) counts were normalized using a sample-specific DESeq2 library size estimation correction factor. Dominant ASVs were determined as ASVs with a relative abundance of 0.001 or higher within a sample. Organic horizon samples are presented in panel A, mineral soil samples are presented in panel B.
(tiff)

**S2 Fig. Ecosystem multifunctionality bacterial diversity (Shannon H and Pielou J) relationship.** Ecosystem multifunctionality (EMF) was calculated by averaging a set of z-score transformed ecosystem functions or nutrient pools. Mineral soil EMF was log-transformed and is displayed without back-transformation. Bacterial diversity was measured using 16S rRNA gene amplicon sequencing. Shaded regions represent a 95% confidence interval. The reported p-values are Benjamini-Hochberg corrected to account for multiple comparisons. In the organic horizon, fall had a slightly higher intercept compared to summer for both Shannon H (A, p = 0.052) and Pielou J (C, p = 0.055). There was no seasonal trend in the mineral soils (B, D). In the organic horizon and the mineral soils, there was no significant relationship between EMF and bacterial Shannon H or between EMF and bacterial Pielou J, as well as no significant differences between the control, warmed for 13 years, and warmed for 28 years treatments.
(tiff)

**S1 Table. Correlation between soil water content and ecosystem multifunctionality and bacterial diversity metrics.** Ecosystem multifunctionality (EMF) and all three bacterial diversity metrics (Shannon H, Chao1 estimated richness, Pielou J) were tested for normality. Variables that were normally distributed (organic horizon Shannon H, Chao1 estimated richness, and Pielou J and mineral soils Shannon H) used a Pearson correlation test, and variables that were not normally distributed (organic horizon EMF and mineral soils EMF, Chao1 estimated richness, and Pielou J) used a Spearman correlation test.
(PDF)

**S2 Table. Non-significant single ecosystem function-diversity relationships in the organic horizon.** All models using a Gaussian distribution and model residuals were checked for

normality. To meet normality of model residuals, β-glucosidase (BG) activity, respiration, and total nitrogen in the Chao1 model were log-transformed. The model estimates and errors for the ecosystem functions that were log-transformed are reported without back-transformation. All model estimates are reported where the reference for warming treatment is control non-heated plots and the reference for season is summer. Number of samples are reported in parentheses. Errors are reported for a 95% confidence interval. Benjamini-Hochberg adjusted p-values are reported. Any adjusted p-values greater than 0.05 are reported as not significant (n.s.).

(PDF)

**S3 Table. Non-significant single ecosystem function-diversity relationships in the mineral soils.** All models used a Gaussian distribution, except total carbon and total nitrogen models, which used an inverse Gaussian distribution with an inverse link function and an inverse Gaussian distribution with an $1/\mu^2$ link function, respectively. Model residuals were examined, and if normality of residuals for the Gaussian models was not met, a log-transformation was applied. Subsequently, N-acetyl glucosaminidase (NAG) was log-transformed to satisfy normality of model residuals. The model estimates and errors for the ecosystem functions that were log-transformed or that used an inverse Gaussian distribution are reported without back-transformation. All model estimates are reported where the reference for warming treatment is control non-heated plots and the reference for season is summer. Number of samples are reported in parentheses. Errors are reported for a 95% confidence interval. Benjamini-Hochberg adjusted p-values are reported. Any adjusted p-values greater than 0.05 are reported as not significant (n.s.).

(PDF)

**S1 Data. Soil organic carbon, soil organic nitrogen, z-score transformed ecosystem functions, and ecosystem multifunctionality indices metadata.** Metadata for the soil organic carbon, soil organic nitrogen, z-score transformed ecosystem functions, and ecosystem multifunctionality CSV file.

(CSV)

**S1 Dataset. Soil organic carbon, soil organic nitrogen, z-score transformed ecosystem functions, and ecosystem multifunctionality indices dataset.**

(CSV)

## Acknowledgments

We would like to thank Jody Potter at the University of New Hampshire Water Quality Analysis Lab for assistance with the total organic carbon and nitrogen analysis.

MSS and KMD conceptualized the study. MSS, LADH, MJC, and RR conducted experiments and collected the data. SDF maintains the long-term soil warming experiments at Harvard Forest, facilitated sample collection, and provided input on data interpretation. MSS and MFM analyzed the data. MSS wrote the first draft of the manuscript. All authors contributed to editing and revising the manuscript.

## Author Contributions

**Conceptualization:** Melissa S. Shinfuku, Kristen M. DeAngelis.

**Data curation:** Melissa S. Shinfuku, Luiz A. Domeignoz-Horta.

**Formal analysis:** Melissa S. Shinfuku, Luiz A. Domeignoz-Horta, Serita D. Frey, Megan F. Mitchell.

**Funding acquisition:** Melissa S. Shinfuku, Serita D. Frey, Kristen M. DeAngelis.

**Investigation:** Melissa S. Shinfuku, Luiz A. Domeignoz-Horta, Mallory J. Choudoir, Ravi Ranjan.

**Methodology:** Melissa S. Shinfuku, Megan F. Mitchell.

**Resources:** Melissa S. Shinfuku, Luiz A. Domeignoz-Horta, Serita D. Frey.

**Software:** Melissa S. Shinfuku, Mallory J. Choudoir, Megan F. Mitchell.

**Visualization:** Melissa S. Shinfuku.

**Writing – original draft:** Melissa S. Shinfuku.

**Writing – review & editing:** Melissa S. Shinfuku, Luiz A. Domeignoz-Horta, Mallory J. Choudoir, Serita D. Frey, Megan F. Mitchell, Ravi Ranjan, Kristen M. DeAngelis.

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
