## [Decision Letter · Decision Letter 0]

10 Jun 2024

PONE-D-24-19172Seasonal effects of long-term warming on ecosystem function and bacterial diversityPLOS ONE

Dear Dr. DeAngelis,

Thank you for submitting your manuscript to PLOS ONE. After careful consideration, we feel that it has merit but does not fully meet PLOS ONE’s publication criteria as it currently stands. Therefore, we invite you to submit a revised version of the manuscript that addresses the points raised during the review process.

The manuscript was thoroughly assessed by two reviewers. Both reviewers find the study interesting, but also express serious concerns, all of which must be addressed before a decision regarding the publication of the manuscript can be reached. Specifically, the reviewers ask for more details about fundamental features of the study site, Reviewer 2 raises a number of critical points regarding the presentation of results and/or statistical evidence as well as the access to original data. Reviewer 1 is questioning the value of aggregating data for multiple factors in a multifunctionality index, a consideration that I share, so please address this comment thoroughly.==============================

We look forward to receiving your revised manuscript.

Kind regards,

Mette Vestergård, Ph.D.

Academic Editor

PLOS ONE

Journal Requirements:

"This work was supported by a National Science Foundation (NSF, https://www.nsf.gov/) Long-Term Research in Environmental Biology grant (DEB-1456528) to SDF and KMD, and an NSF CAREER award (DEB-1749206) to KMD. The soil warming experiments at Harvard Forest are maintained with support from the National Science Foundation (NSF) Long Term Ecological Research Program (DEB-1832110). This work was also supported in part by a Graduate School Dissertation Research Grant from the University of Massachusetts Amherst to MSS (https://www.umass.edu/graduate). This research was also supported [in part] by the intramural research program of the U.S. Department of Agriculture (https://www.usda.gov), National Institute of Food and Agriculture, Hatch Multistate, accession number 7004345. The findings and conclusions in this publication have not been formally disseminated by the U. S. Department of Agriculture and should not be construed to represent any agency determination or policy. None of the funders had any role in study design, data collection and analysis, decision to publish, or preparation of the manuscript."

"We would like to thank Jody Potter at the University of New Hampshire Water Quality 

Analysis Lab for assistance with the total organic carbon and nitrogen analysis. 

This work was supported by a National Science Foundation (NSF) Long-Term 

Research in Environmental Biology grant (DEB-1456528) to SDF and KMD, and an 

NSF CAREER award (DEB-1749206) to KMD. The soil warming experiments at 

Harvard Forest are maintained with support from the National Science Foundation 

(NSF) Long Term Ecological Research Program (DEB-1832110). This work was also 

supported [in part] by a Graduate School Dissertation Research Grant from the 

University of Massachusetts Amherst to MSS and by the intramural research program of 

the U.S. Department of Agriculture, National Institute of Food and Agriculture, Hatch 

Multistate, accession number 7004345. The findings and conclusions in this publication 

have not been formally disseminated by the U. S. Department of Agriculture and should 

not be construed to represent any agency determination or policy. 

MSS and KMD conceptualized the study. MSS, LADH, MJC, and RR conducted 

experiments and collected the data. SDF maintains the long-term soil warming

experiments at Harvard Forest, facilitated sample collection, and provided input on data 

interpretation. MSS and MFM analyzed the data. MSS wrote the first draft of the 

manuscript. All authors contributed to editing and revising the manuscript."

"This work was supported by a National Science Foundation (NSF, https://www.nsf.gov/) Long-Term Research in Environmental Biology grant (DEB-1456528) to SDF and KMD, and an NSF CAREER award (DEB-1749206) to KMD. The soil warming experiments at Harvard Forest are maintained with support from the National Science Foundation (NSF) Long Term Ecological Research Program (DEB-1832110). This work was also supported in part by a Graduate School Dissertation Research Grant from the University of Massachusetts Amherst to MSS (https://www.umass.edu/graduate). This research was also supported [in part] by the intramural research program of the U.S. Department of Agriculture (https://www.usda.gov), National Institute of Food and Agriculture, Hatch Multistate, accession number 7004345. The findings and conclusions in this publication have not been formally disseminated by the U. S. Department of Agriculture and should not be construed to represent any agency determination or policy. None of the funders had any role in study design, data collection and analysis, decision to publish, or preparation of the manuscript."

Additional Editor Comments:

The manuscript was thoroughly assessed by two reviewers. Both reviewers find the study interesting, but also express serious concerns, all of which must be addressed before a decision regarding the publication of the manuscript can be reached. Specifically, the reviewers ask for more details about fundamental features of the study site, Reviewer 2 raises a number of critical points regarding the presentation of results and/or statistical evidence as well as the access to original data. Reviewer 1 is questioning the value of aggregating data for multiple factors in a multifunctionality index, a consideration that I share, so please address this comment thoroughly.

Reviewers' comments:

Reviewer's Responses to Questions

**Comments to the Author**

1. Is the manuscript technically sound, and do the data support the conclusions?

Reviewer #1: Partly

Reviewer #2: Yes

2. Has the statistical analysis been performed appropriately and rigorously? 

Reviewer #1: Yes

Reviewer #2: No

3. Have the authors made all data underlying the findings in their manuscript fully available?

Reviewer #1: No

Reviewer #2: No

4. Is the manuscript presented in an intelligible fashion and written in standard English?

Reviewer #1: Yes

Reviewer #2: Yes

5. Review Comments to the Author

Reviewer #1: I very much enjoyed reading the paper. It is very well-written and developed, and I particularly appreciate the thorough statistical approach and description. However, I see two major issues.

Firstly, I don’t see the value of using a multifunctionality index, as it obscures the individual factors and makes any interpretation in terms of possible mechanisms impossible. Why not use a multivariate approach? Please justify the use of the multifunctionality index in the paper, or consider using a different analytical approach.

Secondly, to enhance the impact of the study, the paper would benefit from the authors explaining why they decided to focus on bacterial diversity only. I miss the context (soil parameters such as nutrient values, soil pH, depth of the organic layer, etc. and a description of the studied forest ecosystem) and a justification for why the impact of the soil fungal communities was disregarded when exploring impacts of the microbial community on the measured ecosystem functions.

Abstract

”We found that ecosystem function was significantly affected by season, with autumn samples having higher function than summer samples.”

Comment: ”higher function” - please specify. Do you mean higher process rates?

”The effect of warming on bacterial diversity was similarly affected by season, where warming in the summer was associated with decreased bacterial evenness in the organic horizon. Despite the decreased bacterial diversity in the warmed plots, we found that the relationship between ecosystem function and bacterial diversity was unaffected by warming or warming duration.”

Comment: These two sentences contain a lot of complex information that is hard to disentangle as is. Instead of using bacterial evenness and bacterial diversity as synonyms (which they aren’t entirely), I would suggest sticking with one or the other. Please specify throughout the paper what you refer to when using the term 'diversity' (total number/evenness).

Also, in the abstract as well as throughout the paper, the fact that the measured ecosystem functions are not determined solely by bacterial diversity needs to be developed. What are the actual soil parameters such as nutrient values, soil pH, etc.? For instance, in nutrient-poor soils, fungi usually play a dominant role in shaping ecosystem functions and comprise a larger proportion of the total microbial biomass compared to bacterial biomass. What are the biomass proportions of these groups in this study? In other words, the lack of a relationship between ecosystem function and bacterial diversity could be explained by bacteria not playing any significant role in the measured ecosystem functions in these soils, and/or diversity (in terms of species numbers) itself not playing a role if only a few taxa actively contribute to the measured ecosystem functions (expressed as a decrease in evenness).

Introduction

Line 27 change “was” to were

Methods

- please add a more detailed description of soil parameters such as, organic layer depth, nutrient content, soil pH, etc. for both sites.

- please add duration between sampling occasion and lab analyses and how samples were treated after sieving in the field

Line 124 please specify “ecosystem functions or soil properties” (x soil properties, x process rates?) I struggle with the term ecosystem functionality because soils are not independent ecosystem but a part of the forest ecosystem

Results

Line 333 What does this mean “In the organic horizon, EMF was significantly higher” given that multifunctionality is a function of both process rates and nutrient pools? If any of the values contributing to EMF change, EMF will change in some way. How can EMF be interpreted without knowing the input values. Can you please add the information of the input values not just their statistical response as presented in Table S1& Table S2

Line 343 what is EMF-diversity?

Discussion

The discussion is comprehensive and well written. The role of the fungal community needs to be addressed throughout the paper.

Line 482 add citation

Reviewer #2: The main aim of the manuscript is to analyze effects of season, soil depth and long-term soil warming on soil bacterial community in temperate deciduous forest using the analysis of relationship between bacterial alpha diversity (assessed by 16S rRNA gene amplicon sequencing) and ecosystem multifunctionality ( using environmental parameters such as enzyme activity, microbial biomass, soil respiration and soil C and N). The manuscript deals with interesting topic and uses the soil warming experimental site which is valuable due to the long-term running and that has been widely studied before. However I have several concerns regarding the manuscript. First of all the authors did not measure soil moisture and soil pH at the studied site, which are very likely the most important environmental parameters shaping microbial communities. Furthermore often results (either data or statistical evidence) that are claimed in the manuscript are not shown- neither in the text nor figures nor tables. Also certain information is reffered to be in specific figure or table , but it is not possible to see it there. Also I miss the access to the original data ( sequence data, enzyme activity or soil properties). See below more specific comments:

Specific comments:

Line 6 … diversity of what?

Line 11-14…. it is true, but there are many studies available that used in-situ climate manipulations to analyze effect of warming on the soil ecosystem (including microbial communitites)

Line 66 … what is meant by active layer of soil? Active layer is the term used in permafrost affected soil

Line 108… what was the diameter of soil cores?

Line 124…. I count 8 different functions, not 7

Line 167-168… you write that organic and mineral horizons were analyzed separatelly, what about samples collected in different seasons?

Line 222-225… explain what issues with sequencing you had…. It is not clear from the description what is meant by the replicates here ? Is it related to original 5 replicates site for each sample?

Line 243-244… what parameters were tested by Wilcoxon and t-test? For t-test were the data normally distributed?

Line 252… soil type is meant soil horizon?

Line 303-304… why only organic horizon and summer samples were tested ?

Line 308….relative abundance treshold is related to entire sequence dataset or individual samples from summer and organic horizon?

Line 309-310… is this related to entire dataset or to summer and organic horizon samples only?

Line 316 … I am unable to see the stated result in Table 1.

Line 327… chao1 increase by 12% is not evident from Table1.

Line 332… can you report any specific results of statistical analysis that would prove such statement?

Line 335…. what means somewhat higher? Be specific, all results you claim has to be supported by statistical analysis which either showes that something was significantly higher or the difference was non-significant

Line 343-344…. in the figure I cannot see any evidence of significant difference that is stated in the text….

Line 369-371… where are these data shown? Where is the evidence for significant changes that are stated?

Fig1 description

A ) Was the EMF from mineral horizon really log transformed as is stated in figure description, if so add this info to the y axis description.

B) the description says that there are p-values reported in the figure- I cannot see them, please explain where is possible to see p-values in the figure.

Fig 2 description

In the figure description explain what is on the figure, do not write the results, they should be obvious from the figure or writen in the result part of the manuscript.

6. PLOS authors have the option to publish the peer review history of their article (what does this mean?). If published, this will include your full peer review and any attached files.

Reviewer #1: No

Reviewer #2: No

---

## [Author Response · Author response to Decision Letter 0]

23 Jul 2024

We would like to thank both the two reviewers and the editors for their time and comments. Please find our response to the comments in the "Response to Reviewers" file. Thank you again.

---

## [Decision Letter · Decision Letter 1]

4 Sep 2024

PONE-D-24-19172R1Seasonal effects of long-term warming on ecosystem function and bacterial diversityPLOS ONE

Dear Dr. DeAngelis,

Thank you for submitting your manuscript to PLOS ONE. After careful consideration, we feel that it has merit but does not fully meet PLOS ONE’s publication criteria as it currently stands. Therefore, we invite you to submit a revised version of the manuscript that addresses the points raised during the review process.

After your thorough revision I am in principle ready to accept the manuscript for publication. However, before my final acceptance, the list of references needs som adjustments to align with the journal format (please see Guide for Authors). Specifically: -Throughout list of references, use journal name abbreviations; please check all references and adjust accordingly -References 5, 13, 27 and 41: Please insert volume/pages/article number==============================

We look forward to receiving your revised manuscript.

Kind regards,

Mette Vestergård, Ph.D.

Academic Editor

PLOS ONE

Journal Requirements:

Additional Editor Comments:

Thanks for your thorough revision of the manuscript, which I am in principle ready to accept for publication.

However, before my final acceptance, the list of references needs som adjustments to align with the journal format (please see Guide for Authors).

Specifically:

Throughout list of references, use journal name abbreviations; please check all references and adjust accordingly

References 5, 13, 27 and 41: Please insert volume/pages/article number

Reviewers' comments:

Reviewer's Responses to Questions

**Comments to the Author**

1. If the authors have adequately addressed your comments raised in a previous round of review and you feel that this manuscript is now acceptable for publication, you may indicate that here to bypass the “Comments to the Author” section, enter your conflict of interest statement in the “Confidential to Editor” section, and submit your "Accept" recommendation.

Reviewer #1: All comments have been addressed

Reviewer #2: All comments have been addressed

2. Is the manuscript technically sound, and do the data support the conclusions?

Reviewer #1: Yes

Reviewer #2: Yes

3. Has the statistical analysis been performed appropriately and rigorously? 

Reviewer #1: Yes

Reviewer #2: Yes

4. Have the authors made all data underlying the findings in their manuscript fully available?

Reviewer #1: Yes

Reviewer #2: Yes

5. Is the manuscript presented in an intelligible fashion and written in standard English?

Reviewer #1: Yes

Reviewer #2: Yes

6. Review Comments to the Author

Reviewer #1: I very much enjoyed working with the authors of this manuscript. It was a constructive and productive process. The responses to all of the reviewers' comments were thorough and thoughtful. I appreciate the discussion about the possible role of the soil fungal community, as well as the honest explanation of why it was not included in the study. I have no further comments and recommend this study for publication in PLOS ONE. Many thanks from me!

Reviewer #2: The authors addressed the raised concerns and answered the questions asked. I would recommend to double check the references, especially capitalization of names of journals should be unified.

7. PLOS authors have the option to publish the peer review history of their article (what does this mean?). If published, this will include your full peer review and any attached files.

Reviewer #1: No

Reviewer #2: No

---

## [Author Response · Author response to Decision Letter 1]

6 Sep 2024

please see specific responses to reviewer comments in the "Response to Reviewers" PDF

---

## [Editor Report · Decision Letter 2]

9 Sep 2024

Seasonal effects of long-term warming on ecosystem function and bacterial diversity

PONE-D-24-19172R2

Dear Dr. DeAngelis,

We’re pleased to inform you that your manuscript has been judged scientifically suitable for publication and will be formally accepted for publication once it meets all outstanding technical requirements.

Kind regards,

Mette Vestergård, Ph.D.

Academic Editor

PLOS ONE
---

## [Editor Report · Acceptance letter]

20 Sep 2024

PONE-D-24-19172R2 

PLOS ONE

Dear Dr. DeAngelis, 

I'm pleased to inform you that your manuscript has been deemed suitable for publication in PLOS ONE. Congratulations! Your manuscript is now being handed over to our production team.

Kind regards, 

on behalf of

Dr. Mette Vestergård 

Academic Editor

PLOS ONE